



# Radiative characteristics of aerosol under smoke mist conditions in Siberia during summer 2012

Tatiana B. Zhuravleva[1], Dmitriy M. Kabanov[1], Ilmir M. Nasrtdinov[1], Tatiana V. Russkova[1], Sergey M. Sakerin[1], Alexander Smirnov[2,3] and Brent N. Holben[3]

[1]V.E. Zuev Institute of Atmospheric Optics SB RAS, Tomsk, Russia

[2]Science, Systems and Applications, Inc., Lanham, MD, USA

[3]NASA Goddard Space Flight Center, Greenbelt, MD, USA

*Correspondence to*: Tatiana Zhuravleva (ztb@iao.ru)

**Abstract.** Microphysical and optical properties of aerosol were studied during mega-fire event in summer 2012 over Siberia using ground-based measurements of spectral solar radiation at AERONET site in Tomsk and satellite observations. The data were analyzed using multiyear (2003-2013) measurements of aerosol characteristics under background conditions and for less intense fires, differing in burning biomass type, stage of fire, remoteness from observation site, etc. ("ordinary fires"). In June – August 2012, the average aerosol optical depth (AOD, 500 nm) had been 0.95±0.86, about a factor of 6 larger than background values (0.16±0.08), and a factor of 2.5 larger than in "ordinary smokes. The AOD values were extremely high on July 24-28 and reached 3-5. Comparison with satellite observations showed that ground-based measurements in the region of Tomsk not only reflect the local AOD features, but also are characteristic for the territory of the Western Siberia as a whole. Single scattering albedo (SSA, 440 nm) in this period ranged from 0.91 to 0.99 with the average of ~0.96 in the entire wavelength range of 440-1020 nm. The increase in absorptance of aerosol particles (SSA(440 nm)=0,92) and decrease in SSA with wavelength, observed in "ordinary smokes", agree with the data of multiyear observations in analogous situations in boreal zone of USA and Canada. Volume aerosol size distribution in smoke mist and ordinary smokes had bimodal character with significant prevalence of fine mode particles, but in summer 2012 the mean median radius and the width of the fine mode distribution somewhat increased. In contrast to data of multiyear observations, in summer 2012 an increase in the volume concentration and median radius of the coarse mode was observed with the growing AOD.

The calculations of the "average" radiative effects of smoke and background aerosol are presented. As compared to background conditions and "ordinary smokes", under the conditions of smoke mist the cooling effect of aerosol considerably intensifies: direct radiative effects (DRE) at the bottom (BOA) and at the top of the atmosphere (TOA) are -13, -35, and - 60 W m$^{-2}$ and -5, -14, and -35 W m$^{-2}$ respectively. The maximal values of DRE were observed on July 27 (AOD(500 nm)=3.5), when DRE(BOA) reached -180 W m$^{-2}$, while DRE(TOA) and DRE of the atmosphere were -80 W m$^{-2}$.



*Key words*: Sun-sky radiometer, AERONET, smoke mist in Siberia 2012,  optical and microphysical
properties of aerosol particles, diurnally radiative effects of smoke and background aerosol

**1 Introduction**

Massive forest and peat fires are the strongest source of aerosol-gas emissions on the territory of boreal zone
of the Northern Eurasia. In addition to anthropogenic factor, the biomass ignition is favored by dangerous
consequences of the global climate warming, manifesting in the form of strong temperature, circulation, and
hydrologic anomalies (Groisman et al., 2007). In view of enormous stretches of boreal zone, fires occur in
some or another region of this vast territory almost every year and influence strongly the radiation budget, air
quality, human health, biological diversity, glaciology, etc. To understand the effects of biomass burning on
the atmosphere, the physical, chemical and optical properties of smoke particles, it needs to be studied and
parameterized with reliable uncertainties.

Since 1990s, a large amount of information on characteristics of carbonaceous particles based on in situ

measurements and remote sensing using the ground-, aircraft- and satellite-based instruments has been
accumulated in hundreds of manuscripts. However, even with the availability of such a volume of data, the
determination of key parameters for estimating atmospheric effects of biomass burning is not straightforward,
primarily because the smoke properties strongly depend on the set of a variety of reasons, the main of which
are the type of biomass, the stage of fire, meteorological conditions at the fire site and on the territory of the
dispersal of smoke plumes, age of smoke, etc. (see, e.g., the reviews Dubovik et al., 2002; Reid et al., 2005,
2005a; Bond and Bergstrom, 2006; Moosmüller et al., 2009; Giles et al., 2012; Sayer et al., 2014; Nikonovas
et al., 2015 and bibliography therein). Another problem is that specific features of individual fires may
strongly differ from the ensemble smoke hazes that are a result of some averaging procedures of characteristics
of numerous fires. This aspect is crucial for studying the radiation effects of aerosol because, in the framework
of regional and global climate models, the most important issue is the development of model representations
concerning the aged smoke that dominates regional hazes and affects climate.

A stable anticyclone had formed in summer 2012 in Siberian region under the conditions of small-

gradient high-pressure baric field (Polyakov et al., 2014), with consequences being that forest and peat fires
burned in a few regions of Siberia and encompassed the territory from 1 to 10 million hectares, by different
estimates. Optical and microphysical properties of near-ground aerosol and specific features of their vertical
structure according to data of in situ measurements, as well as spatiotemporal evolution of aerosol optical
depth and active fires according to results of satellite monitoring, are presented in (Gorchakov et al., 2014;
Kozlov et al., 2014; Sklyadneva et al., 2015; Vinogradova et al., 2015; Panchenko et al., 2016). In our work,
we discuss the columnar optical and microphysical aerosol characteristics, retrieved on the basis of ground-
based photometric observations in Tomsk during the period of smoke mist in 2012. The second aim is to
compare these results with data of multiyear photometric measurements and satellite observations over the
territory of the Western Siberia under the different atmospheric conditions.



## 2 Instrumentation, Sites, and Methods

### 2.1 Ground-based measurements

The measurements, presented in this paper, were made with CIMEL Electronique CE 318 sun-sky radiometer that is a part of AErosol RObotic NETwork (AERONET, (Holben et al., 1998)). These measurements were performed in eastern suburb of Tomsk ("Tomsk": 56.48°N; 85.05°E) in period of 2003-2010; and since 2011, CE 318 has been operated at "Fonovaya" observatory located 60 km away from the city ("Tomsk-22": 56.42ºN; 84.07ºE).

Direct Sun measurements are made in the spectral channels centered at 340, 380, 440, 500, 675, 870, 940 and 1020 nm (bandwidth: 10 nm at full width at half maximum). These solar extinction data are then used to compute aerosol optical depth (AOD, $\tau_\lambda$) at each wavelength $\lambda$ except 940 nm, which is used to retrieve total columnar content of water vapor ($W$). The spectral aerosol optical depth data have been screened for clouds following the methodology of Smirnov et al. (2000). In addition to direct Sun, radiation measurements in solar almucantar are made in four channels of CE 318: 440, 675, 870, and 1020 nm. These data and the inverse automated algorithm of Dubovik and King (2000) (version 1) with enhancements (version 2, Holben et al., 2006) were used to retrieve other aerosol characteristics: volume particle size distribution function $dV/d\ln r \left(\mu m^3 \ \mu m^{-2}\right)$, scattering phase function, asymmetry factor $g_\lambda$, complex refractive index ($n_\lambda - \kappa_\lambda \times i$), and single scattering albedo $\omega_\lambda$.

Dubovik et al. (2000a) presented uncertainty of retrieval estimates for volume size distribution, refractive index, and single scattering albedo (SSA). For Level 2 data at moderate aerosol loading ($\tau_{440}$ ~0.4) uncertainties for such retrievals are: 10-35% for the binned size distribution in the intermediate particle size range ($0.1 \le r \le 7\,\mu m$); 0.04 and 30-50% for the real and imaginary parts of the refractive index; and 0.03 for single scattering albedo. For typical biomass burning models (Dubovik et al., 2002) these uncertainties were propagated onto the other size distribution parameters (Sayer et al., 2014): the volume mean radius $r_v$ and standard deviation σ of fine (f) and coarse (c) modes are retrieved with errors 0.01 µm for $r_v^f$, 0.2 µm for $r_v^c$, and 0.06 for $\sigma^f$ and $\sigma^c$. This led to uncertainties of ~0.015–0.04 in the asymmetry factor (AF) of fine aerosol mode (larger uncertainties at longer wavelengths) and ~0.01 in the coarse mode aerosol (smaller uncertainties at longer wavelengths).

An original approach, relying upon ground-based spectral measurements of AOD and radiance phase functions, was also used in addition to the algorithm of Dubovik and King (2000) to solve the inverse problem. The first version of the algorithm, namely, Sun-Sky Measurements for Aerosol ReTrieval (SSMART 1.1 software package), was implemented under the assumption of (a) the sphericity of aerosol particles and (b) the independence of the complex refractive index on the wavelength and particle size (Bedareva et al., 2013a). An improved version of the algorithm (SSMART 1.2), in which the inverse light scattering problem was solved using the model of a mixture of randomly oriented polydisperse spheroids, was suggested by Bedareva et al.



(2014). In both versions, the aerosol optical characteristics are derived in two ways: directly from the spectral
Sun-sky radiometer measurements (Way 1) and through the calculation based on the retrieved particle size
distribution and complex refractive index (Way 2). On the basis of closed numerical experiments, the accuracy
of aerosol retrievals was investigated in error-free conditions and in the presence of measurement errors. The
SSMART algorithm was tested as applied to conditions of moderate and increased aerosol turbidity of the
atmosphere at Tomsk and Dakar (14 N, 16 W) AERONET sites. It was found that the SSMART and
AERONET codes give the consistent estimates of aerosol properties within their retrieval uncertainties
(Bedareva et al., 2013ab, 2014).
Measured aerosol optical depth and computed retrieval products were used to derive additional aerosol
properties. Spectral dependence of AOD is traditionally described by the empirical Ångström formula

$$\tau_\lambda = \beta\lambda^{-\alpha}. \tag{1}$$

The absorption AOD (AAOD) is calculated for each wavelength using the following equation (Giles et al.,

2012)

$$\tau_{abs,\lambda} = \tau_\lambda \times \left[1 - \omega_\lambda\right] \tag{2}$$

and, similar to Eq. (1), it can be represented as

$$\tau_{abs,\lambda} = \beta_{abs}\lambda^{-\alpha_{abs}}. \tag{3}$$

Analyzing the radiative characteristics of aerosol in the period of 2012 smoke mist, we additionally
employed data of multiyear AOD observations at "Tomsk" and "Tomsk-22" sites, obtained from April to
October in 2003-2011 and in 2013. Using the method of Kabanov and Sakerin (2006), the total dataset was
divided into two subsets: 1) "background" (usual) conditions (998 days); and 2) "ordinary" smoke situations
(81 days). By the "ordinary" smokes we mean smoke situations from yearly observed Siberian biomass
burning of different types (forest and peat fires, springtime vegetation burning, smokes from remote sources),
which were shorter lasting and less severe as compared to 2012 smoke mist. Simultaneous measurements at
these sites revealed no statistically significant AOD differences in the warm period of the year (Sakerin et al.,
2010); therefore, merging of data, obtained at the neighboring sites ("Tomsk" and "Tomsk-22"), can be
regarded as correct.
Smaller data volume had gone into comparative analysis of aerosol optical and microphysical
characteristics obtained from the inverse procedure. The number of retrievals of total set of characteristics,
including refractive index, single scattering albedo, volume size distribution function, and scattering phase
function (Level 2, $\tau_{440} > 0.4$), had been 65 in the period of smoke mist (June 15 – August 10, 2012), and 140
under the conditions of "ordinary" smokes (April – October 2003-2011, 2013). Out of 140 "ordinary" smoke
situations, maximal number of almucantar retrievals had been 39 (2004), 23 (2006), and 22 (2013). In these
data, results obtained in May 2004 deserve a special attention. Dry and warm weather, predominating in
Novosibirsk and Tomsk regions (with air temperatures exceeding 30ºC on separate days) led to numerous
forest fires, provoked by burning of last year's vegetation and bonfires, left unextinguished by fishers and
hunters. The number of retrieved aerosol characteristics varied from 1 (2005) to 15 (2008) for remaining
period of multiyear observations.





**2.1 Satellite data**

AOD observations at 550 nm (collection 6, http://giovanni.sci.gsfc.nasa.gov/giovanni/) and Aerosol Small Mode Fraction (collection 5, http://gdata1.sci.gsfc.nasa.gov/daac-bin/G3/gui.cgi?instance_id=MODIS_DAILY_L3, valid at time of manuscript preparation) from MODIS instruments were used. These MODIS products (TERRA and AQUA platforms, Level 3 data, i.e., daily averaged within the grid cells 1º×1º) were obtained using the algorithm of Levy et al. (2010).

**3 General characteristics of the large-scale smoke pollution in the summer of 2012**

In this section, we present the general characteristics of smoke situation in 2012 on the territory of Tomsk region.

**3.1 Weather-climate features**

In summer 2012, a stable anticyclone had formed over the territory of Siberia under the conditions of small-gradient high-pressure baric field (Polyakov et al., 2014), the consequences being substantial changes in climatically significant characteristics and, primarily, an appreciable increase in air temperature and a decrease in precipitation. Data of observations at "Tomsk" meteorological station (WMO_ID=29430, http://rp5.ru) indicate that monthly mean temperature in July was maximal over last decade, while precipitation amount was close to a minimum (Figure 1). On the whole, positive anomalies of temperature reached 1.3-7.2ºC in June-July on the territory of Tomsk region (~56-61ºN; 75-88ºE), and precipitation amount was 20-30% of climatic norm. Selyaninov hydrothermal wetting coefficient was also used as a characteristic of wetting (drought) regime (see, e.g., (Polyakov et al., 2014)); it is defined as the ratio of precipitation total for period no shorter than one month to the sum of temperatures for the same period, decreased by a factor of ten. Polyakov et al. (2014) showed that, over the last 70 years, the atmospheric droughts on the territory of Tomsk region were longer than one month in nine summer seasons (the hydrothermal wetting coefficient varied from 0.6 to 1.4), with the summer of 2012 being the driest one.

These conditions had led to extensive forest fires in the Western and Eastern Siberia (Figure 2), accompanied by considerable pollution of the atmosphere by combustion products (smoke particles, carbon and nitrogen oxides, etc.). Ground-based observations in Tomsk indicate  that, in period of maximal smoke pollution on July 25-28, 2012, the aerosol number concentration had been 3000-8500 cm$^{-3}$ (particle size 0.4-15 μm), exceeding the background values by about a factor of 20, while carbon monoxide concentrations reached 7.7 ppm (Sklyadneva et al., 2015). Results of satellite (AIRS/TERRA) monitoring indicate that the July-average total CO content over the territory of Tomsk region in 2012 increased by approximately 40% (Fig. 1) as compared to 2005-2015 (except 2012).





Anticyclone over the territory of the Western Siberia persisted for almost two months (second half of
June – first decade of August), similar to anomalous situations observed over the European territory of Russia
in 1972 and 2010 (Shakina and Ivanova, 2010). At the end of first decade of August, the blocking cyclone
started to break, heat rapidly weakened, and air temperature returned to within the climatic norm at all
meteorological stations of Tomsk region (Polyakov et al., 2014).

**3.2 Aerosol radiation characteristics**
**3.2.1. Ground-based observations**

Figure 3abc shows the time variations in daily average values of AOD, single scattering albedo $\omega_{440}$, and
asymmetry factor $g_{440}$ in the summer period of 2012.
The AOD data indicate that there are two waves of high atmospheric smoke turbidities: from June 17 to
July 5, and from July 19 to August 6. Maximal AOD values (500 nm), reaching 3-5, were observed on July 3-6
and July 25-29.
The single scattering albedos did not go out of the interval $0.93 \le \omega_{440} \le 0.99$ in almost all cases. The
asymmetry factor $g_{440}$ varied from 0.66 to 0.74, i.e., in the same range as multiyear average values on the
territory of Siberia (Sakerin et al., 2009, 2012, 2014). From results presented here it also follows that, on those
days when SSA and AF were retrieved by two methods, the SSMART and AERONET codes gave consistent
estimates of aerosol properties within their retrieval uncertainties.
Arrival of smoke plumes was also accompanied by a substantial increase in the mass concentrations of
black carbon ($M_{BC}$) and aerosol ($M_a$) in the near-ground atmospheric layer. Kozlov et al. (2014) showed that
the average values of these characteristics at "Fonovaya" observatory in the period from June 17 to August 4
varied in the ranges $M_{BC}$=1.9-10.3 µg m$^{-3}$ and $M_a$=103-425 µg m$^{-3}$ given the background levels of ~0.45 and
~20 µg m$^{-3}$ respectively for August 10-31. Similar to AOD, the maximal concentrations of $M_{BC}$ and $M_a$ were
observed in the period of July 25-29.
The examples of daily average aerosol volume size distribution $dV/d\ln r$ for moderate to high AOD
values are presented in Figure 3d. A noticeable salient feature of the $dV/d\ln r$ distribution is the increase in
the modal radius and in the width of the mode of finely dispersed fraction during extreme turbidity on July 28
as compared to periods on June 27 and July 14, when AOD was much lower.
A consequence of anomalously high atmospheric turbidities had been substantial changes in solar
radiation reaching the earth's surface. Figure 4 illustrates the daytime  behavior of direct and diffuse radiative
fluxes, measured with MS-53 pyrheliometer and MS-802 pyranometer (0.305-2.8 µm) under usual
(background) conditions on July 24, 2011 ($\tau_{500} = 0.07$) and in the highly turbid atmosphere on July 28, 2012
($\tau_{500} = 2.14$). Under the influence of smoke plume, there was a two- to four-fold decrease in direct radiation,
which was partially compensated for by approximately equal increase in diffuse radiation. The diffuse





radiation was considerably (a factor of ~1.8) larger than the direct radiation throughout the day (July 28,
2012).
**3.2.2 Satellite observations**
The spatial distributions of AOD at 550 nm and Aerosol Small Mode Fraction ( $\eta_{550} = \tau^{f}_{550} / \tau_{550}$ ) over
Western and partially Eastern Siberia (60º-100ºN, 52º-70ºE), averaged over the period of June 15 – August 10,
2012, are presented in Figure 5. These data show that the territory of the Tomsk region in this period was
subjected to intensive influence of large-scale smoke pollution and characterized by high values of AOD and
content of fine aerosol fraction: $\tau_{550} > 0.9$ ; $\eta_{550} > 0.7$ .
**4.        Discussion of results**
**4.1       Temporal and Spectral Variability of Aerosol Optical Depth**
**4.1.1.    Ground-based data**
Analysis of multiyear AOD variations in a few regions of Russia according to data of photometric
observations showed that, after eruption products of Pinatubo volcano had sunk out of the stratosphere, the
interannual AOD variations were small and no statistically significant trend was observed in the past two
decades (Sakerin et al., 2009, 2012; Sakerin and Kabanov, 2015). For instance, the annually average AOD
values (500 nm) in Tomsk during the period of 2003-2015 (except in 2012) varied from 0.13 to 0.21
(Figure 6a).
Under the influence of massive forest fires in Siberia during summer 2012, the annual AOD value had
exceeded the multiyear norm by almost a factor of two and became the highest over the entire period of
photometric observations in Tomsk (since 1992) (Sakerin and Kabanov, 2015). Average summertime AOD
values changed even stronger: in particular, the average value $\tau_{500}$ in July had increased by a factor of 4.5 as
compared to multiyear data (Figure 6b). Even if we omit anomalously high turbidities ( $\tau_{500} > 1$ ), which may
be partly due to clouds invisible through haze, the average AOD values in summer months of 2012 go out of
the corridor defined by standard deviations (SDs).
More detailed characteristics of atmospheric AOD for three wavelengths in UV, visible, near IR ranges
during  the smoke mist in comparison with the multiyear average data for July are presented in Table 1. Also
given are the average characteristics for various situations of "ordinary" smokes, which are observed every
year in Siberia in warm period. The comparison showed (see also Figure 6c) that AOD during smoke mist
exceeds the background values by a factor of 5.5-6 in the entire spectral range. In absolute value, AOD
increased stronger in shortwave part of spectrum, i.e., due to finely dispersed aerosol. The slight increase in
Ångström exponent α, which depends on the interrelation between contributions of fine and coarse aerosols to



AOD, also indicates that small particles predominate in smoke aerosol. Despite the temperature, aerosol, and
other anomalies, the total water vapor content of the atmosphere in the period of smoke mist differed little
from the multiyear norm.

Average AOD characteristics in "ordinary" smokes incorporate data for fires of different types and

distances from the region of measurements and, as such, occupy an intermediate position between background
conditions and smoke mist. Independent of their intensity (mist or "ordinary" smokes), a common feature of
all smokes is an identical exponent α, which characterizes higher content of fine aerosol.

It is noteworthy that the multiyear average values of α and AOD in the near-IR range, calculated for

background conditions and total dataset, differ little (see columns 2 and 3 in Table 1), primarily because of
relatively small number (about 8%) of smoke situations in the total dataset.

Under the assumption that aerosol size distributions are bimodal, O'Neil et al. (2001, 2003) have

developed a spectral deconvolution algorithm (SDA) to infer the component fine and coarse mode optical
depth from spectral dependence of AOD. In June-August 2012, the coarse mode $\tau_{500}^{c}$ is less than fine mode
$\tau_{500}^{f}$ and is typically low (only 3 days show that $\tau_{500}^{c} \sim 0.2$), whereas $\tau_{500}^{f}$ is high and exhibits very large day-
to-day variability (Figure 7a). The parameter $\eta_{500}$, which characterizes the contribution of fine fraction to
AOD at 500 nm, exceeded 0.7 and approached 0.9 at $\tau_{500} \geq 0.6$ (Figure 7b). These results are consistent with
earlier published data, according to which content of finely dispersed aerosol predominately increases during
vegetation burning in the absence of any other significant sources (Reid et al., 2005a; Eck et al., 2009; Giles et
al., 2012).

**4.1.2 Analysis of satellite observations**

Analysis of spatiotemporal AOD variations was confined to the consideration of the central part of the
Western Siberia. A salient feature of this territory is a uniform landscape (without mountainous terrain) and
the absence of large sources of anthropogenic pollution. Under usual conditions, the AOD values, retrieved
from satellite observations, are characterized by small (background) values and by quite a uniform spatial
distribution (Zhuravleva et al., 2009a; Sakerin et al., 2012).

Since the formation and evolution of smoke plumes are strongly affected by large-scale atmospheric

dynamics, spatial distributions of AOD may substantially change from one day to another. The main specific
features of variations can be identified through analysis of AOD fields, averaged for a few 5-day periods,
selected by taking into account the ground-based observations (Fig. 3a): on June 6-10 there were background
conditions (before beginning of forest fires); on July 1-5 and July 24-28 there were maximal smoke turbidities;
and on July 10-14 there were relatively low AOD values observed between two turbidity maxima. Hereinafter,
for   calculations of the AOD values averaged over the selected time interval (5 days) we used the daily
satellite product.

Spatial AOD distributions, presented in Figure 8, show that the "Tomsk-22" observation site was either

at the center (Fig. 8b) or on the periphery of smoke plumes (Fig. 8a) in different periods of time. The periods





of small turbidities of the atmosphere (on June 6-10 and July 10-14) were characterized by quite a uniform
AOD distribution, and the largest spatial inhomogeneities and values $\tau_{550} > 1$ were observed over the central
part of the Western Siberia on July 24-28 (Fig. 8 and Table 2).

The statistical characteristics in Table 2 were calculated using measurements from AQUA and TERRA

platforms (MODIS collection 6, http://giovanni.sci.gsfc.nasa.gov/giovanni). These data suggest that, for three
out of four periods considered here, the average AOD values at "Tomsk-22" site are close to spatiotemporal
averages on the entire territory of the Western Siberia: the difference between satellite and ground-based data
is much less than standard deviations, except in period of July 24-28, when the average AOD at "Tomsk-22"
site had been $\tau_{550} = 2.74$ and exceeded the average value for the Western Siberia (MODIS) $\tau_{550} = 1.06$. At
the same time, the satellite data in the (1°×1°) region of "Tomsk-22" site well agree with results of photometric
observations. Thus, the results of ground-based measurements in the region of "Tomsk-22" can be considered
to reflect not only the local AOD features, but also the regularities of variations for the entire territory of the
Western Siberia.

**4.2. Retrieval results**

We will compare the retrievals of aerosol optical and microphysical characteristics under the conditions of
smoke mist in 2012 against the average data for ordinary smokes (Table 3). Since the AOD dependence of $C_v$
and $r_v$ for fine and coarse aerosol modes has previously been noted for a wide variety of aerosol types
(including biomass burning aerosol (Dubovik et al., 2002; Sayer et al., 2014)), Table 3 also presents the
resulting linear regression relationships between the pairs ($C_v^{f(c)}, \tau_{440}$) and ($r_v^{f(c)}, \tau_{440}$), as well as the
corresponding linear correlation coefficients $R$.

**4.2.1 Volume size distributions**

During the period of smoke mist in 2012, the average values of volume median radii for finely and coarsely
dispersed fractions were $r_v^f = 0.18\,\mu m$ and $r_v^c = 3.3\,\mu m$, a little larger than these characteristics for ordinary
smokes, equaling 0.16 µm and 2.9 µm respectively (Table 3). The $r_v^f$ value, in both cases, was in the range of
values, obtained for biomass burning in other regions of the globe (Reid et al., 2005, 2005a; Eck et al., 2009;
Chubarova et al., 2012), as well as in the range of measurements for aged smoke, performed using Optical
Particle Counter $(0.1 - 3\,\mu m)$ and Differential Mobility Particle Sizer $(0.01 - 0.6\,\mu m)$ in boreal zones of
Europe and North America (Reid et al., 2005). Like in Siberia, the wider size distribution of fine particles than
for ordinary smokes (Table 3; Fig. 3d) was also noted by other authors  for severe fires in boreal zone of
Alaska (Bonanza Creek, 2004-2005, Eck et al., 2009).

The noteworthy salient features of finely dispersed fraction during summer 2012 are as follows. The

contribution of finely dispersed fraction to the total volume ($C_v^f / C_v^t$), on the average, increased to



$0.8 \pm 0.065$ as compared to value $0.65 \pm 0.21$ according to multiyear observations. Volume concentrations and
volume median radii of the fine mode are plotted in Figure 9a as functions of AOD at 440 nm. This figure
shows that there is general increase in volume concentrations $C_v^{\mathrm{f}}$ as $\tau_{440}$ increases (Fig. 9a). The median
radius of the fine mode increased from about 0.15 to 0.22 µm as $\tau_{440}$ changed from 0.4 to 1.5 and larger (Fig.
9b). The smoke particles have large radii, and $r_v^{\mathrm{f}}$ and $\tau_{440}$ are closely correlated, possibly because of high
concentration of aerosol, which leads to greater coagulation, condensation, and gas-to-particle conversion
(Reid et al., 1998, 2005; Eck et al., 2009). From this figure it also follows that the interrelation between
median radius of finely dispersed fraction and AOD at 440 nm in ordinary smokes is much less pronounced,
primarily because of the variety of properties biomass burning aerosol combined in this dataset. For instance,
in May 2004, the median radius was relatively small ($r_v^{\mathrm{f}}$ ~0.14 µm) and did not practically change with
varying $\tau_{440}$. Most probably, this was because burning of last year's vegetation was one of the sources of
smoke particles in this period of time. (An analogous feature was also observed during African savanna fires
(Dubovik et al., 2002)).

In contrast to data of multiyear observations, in summer 2012 an increase in the volume concentration

and median radius of the coarse mode was observed with the growing AOD. Most probably, soil particles,
being suspended by saltation of surface dust driven by fire generated winds, were also present during this
period of time in the composition of coarse mode in addition to carbon aggregates.

Results of retrieval of disperse composition of finely dispersed aerosol in the atmospheric column agree

with measurements in near-ground layer (Kozlov et al., 2014). Data of spectral-polarization nephelometric
measurements were used to show that the volume median radius of the fine mode under the conditions of weak
turbidity of the atmosphere (July 10–13, 2012) had increased from 0.1 µm to 0.4-0.5 µm when smoke plumes
intruded to the region of observations (July 25–29, 2012).

**4.2.2 Refractive index**

Average values of the real and imaginary parts of refractive index under the conditions of smoke mist and
ordinary smokes are presented in Table 3. In ordinary smokes, the imaginary part of refractive index $\kappa_\lambda$
changes weakly with the increasing wavelength and is approximately 0.01. However, the imaginary refractive
index in the Tomsk smoke mist shows low values and relatively large decrease in $\kappa_\lambda$ as the wavelength grows
from 440 to 675 nm. The $\kappa_\lambda$ variations in the wavelength interval of 675-1020 nm are not as strong.

The quantitative differences and specific features of spectral dependence of the imaginary refractive

indices stem from different properties of atmospheric carbonaceous particles, i.e., black carbon (BC) and
organic aerosol (OA). Recent studies showed that OA components can contribute substantially to light
absorption. In contrast to BC, which absorbs light throughout the UV-visible spectrum, such an OA
component as "brown carbon" (BrC) absorbs mostly in the ultraviolet wavelength and less significantly in the
visible spectral range (Kirchstetter et al., 2004; Bergstrom et al., 2007; Chen and Bond, 2010; Zhong and



Jang, 2014). The majority of BrC is emitted to the atmosphere through low temperature, incomplete
combustion of biomass, bio- and fossil fuel (Bond, 2001; Kirchstetter et al., 2004; Bergstrom et al., 2007;
Lewis et al., 2008; Chen and Bond, 2010; Zhong and Jang, 2014).
The absorption efficiency and spectral dependence vary depending on the type of BrC origin. Lu et al.
(2015) reviewed available measurements (laboratory and field observations) of light-absorbing primary
organic aerosols, and quantify the wavelength-dependent imaginary refractive indices ($\kappa_{OA}$) for the bulk
primary OA emitted from biomass/biofuel, lignite, propane and oil combustion sources. Based on generalized
information, they suggested to parameterize $\kappa_{OA}$ of biomass/biofuel combustion sources as a function of BC-
to-OA ratio. Analysis of imaginary refractive indices showed the stronger wavelength-dependent $\kappa_{OA}$ for
lower BC-to-OA ratio conditions (smoldering combustion), while the greater $\kappa_{OA}$ values at $\lambda > 350$ nm are
observed for higher BC-to-OA ratio (flaming combustion).
These results allow us to hypothesize that the reason for the $\kappa_\lambda$ decrease in the interval of 440-675 nm
during summer 2012 may be due to the contribution of compounds absorbing radiation in UV wavelength
region ("brown" carbon). To confirm this hypothesis, we considered OMI observations of the Ultraviolet
Aerosol Index (UVAI), the value of which is sensitive to aerosol absorption in the ultraviolet wavelength
region (Torres et al., 1998; Jethva and Torres, 2011; Hammer et al., 2016)]. Satellite data indicate that UVAI
values were quite high in the period of smoke mist, and exceeded 4-5 on individual days, thus explaining the
above-mentioned $\kappa_\lambda$ decrease in going from UV to visible spectral region (Figure 10).
The relatively constant imaginary refractive index in ordinary smokes owes to averaging over many
situations, differing, in particular, in the type of burning biomass (peat, forest, grass). For instance, in May
2004, when the characteristics of smoke aerosol were quite homogeneous, the average spectral behavior of $\kappa_\lambda$
was weakly manifested, with $\kappa_\lambda \approx 0.013$ in the interval of 440-1020 nm. At the same time, the UVAI value
did not exceed 1.2 (http://giovanni.sci.gsfc.nasa.gov/giovanni/), suggesting that BC was seemingly the main
absorbing substance in this period of time.

**4.2.3 Single scattering albedo**

Detailed analysis of spectral SSA in 2012 has revealed two types of dependence: monotonically decreasing
and monotonically increasing $\omega_\lambda$ with the growing wavelength (Figure 11).
The main group of SSA data (45 cases out of 65) is characterized by a weakly decreasing spectral
dependence: the range of the differences $\Delta\omega_\lambda = \omega_{440} - \omega_{1020}$ did not exceed 0.03 (Fig. 11a). Practically all set
of retrieved SSA of this type lies between the curves 2 ($\omega_{440} = 0.926$) and 3 ($\omega_{440} = 0.996$). In other cases,
the $\omega_\lambda$ decrease with the growing wavelength is larger, but does not exceed 0.06 (curve 4, Fig. 11a). We note
that the values of the imaginary part of refractive index are ~0.006 for this subset of data.





In a few (14 out of 65) situations with high atmospheric turbidities ($\tau_{440} > 0.9$), there was an increase
in SSA with the growing wavelength (curve 1 in Fig. 11a). The monotonic increase in $\omega_\lambda$ agrees with
decrease in $\kappa_\lambda$ (on the average, from 0.01 at 440 nm to ~0.006 at 675 nm), presumably due to additional
absorption of "brown" carbon in the UV wavelength region.
On the whole, the 2012 smoke mist was characterized by weak spectral variations in SSA, the average
being ~0.96, consistent with previous results in other regions of Eurasian boreal zone: in Moscow (2002 and
2010) and Alaska (2004-2005), where the smoldering phase prevailed over the flaming fire phase (Eck et al.,
2009; Chubarova et al., 2011; Sayer et al., 2014). The relatively constant spectral single scattering albedo of
smoke mist in 2012 may be also partly due to growth of aerosol particle sizes (Subsect. 4.2.1, Table 3; see also
Eck et al., 2009).
The SSA retrievals in atmospheric column satisfactorily agree with data in the ground-layer (Kozlov et
al., 2014), according to which the single scattering albedos in the visible range ($\lambda$=510 nm) were in the interval
of 0.95-0.98. At the same time, the imaginary and real parts of the complex refractive index of dry base of
substance varied in the ranges of 0.01-0.02 and 1.36-1.47 respectively.
Overwhelming majority of results of individual retrievals in "ordinary" smokes also showed a
monotonic SSA decrease with the increasing wavelength, with the opposite dependence being observed in
only 6 cases out of 140. However, in contrast to smoke mist, the spectral SSA behavior was stronger
pronounced: $\Delta\omega_\lambda$ reached 0.06-0.18 in 18% of cases; $\Delta\omega_\lambda$ was 0.03-0.06 in 40% of cases, and $\Delta\omega_\lambda$ was
less than 0.03 in 34% of cases.
The average single scattering albedos in "ordinary" smokes, presented in Fig. 11b, turned out to be
about 0.02 lower than those observed in boreal forests of USA and Canada (Dubovik et al., 2002). These
differences may be both due to methodic and physical factors. Methodic-type differences might stem from
different methods for selecting the smoke situations. The specific features of the spectral dependence of SSA
can also be explained by different combinations of the composition and age of smoke aerosol, which were
recorded at AERONET observation sites. This is clearly illustrated by results obtained in May 2004 in Tomsk
(Fig. 11b). Lower values and pronounced spectral variations of SSA (a decrease from 0.9 to 0.85) may be due
to the last year's grass and agricultural vegetation combustion products, present in the smoke composition, as
well as due to flaming phase of combustion which predominated in that period of time. (Close values and
analogous spectral dependence of single scattering albedo were also noted in African savanna fires (Zambia),
Dubovik et al., 2002, Sayer et al., 2014).
Aerosol absorption optical depth $\tau_{abs,\lambda}$ and absorption Ångström exponent $\alpha_{abs}$, computed for the
spectral interval of 440-870 nm (Sect. 2), were considered as other characteristics of absorbing properties of
aerosol particles.
Data, presented in Fig. 11d, show that the average values of $\tau_{abs,\lambda}$ in the period of smoke mist were
about a factor of 1.5-2 smaller than for "ordinary" smokes. In addition, there are differences in $\alpha_{abs}$, which
are 1.6±0.44 (smoke mist) and 1.2±0.5 (ordinary smoke) respectively.





The AAE values can be used as an indicator of aerosol composition (Andreae and Gelencsér, 2006;
Russell et al., 2010; Schuster et al., 2016). The relatively low values $\alpha_{abs} \sim 1$ are typical for aerosol absorption
largely dominated by black or light absorbing carbon, while larger values suggest absorption by different
material organic carbon (Bergstrom et al., 2002, 2007; Kirchstetter et al., 2004; Lewis et al., 2008; Russell et
al., 2010; Schuster et al., 2016). The $\alpha_{abs}$ increase in the 2012 fires may indicate that, on the average, smoke
absorption possibly shifted from domination by black carbon in "ordinary" smokes to an increasing influence
of absorption by other materials most likely organic carbon (see also Fig. 10).
In contrast to single scattering albedo, the spectral asymmetry factor differs insignificantly among the
2012 smoke mist, "ordinary" smokes, and May 2004 smokes (Table 3, Figure 11c).

**5. Radiative effects of smoke aerosol**

The radiative effects of smoke aerosol were accounted for by calculating the transmittance ($T$), albedo ($A$), and
absorptance of the atmosphere ($ABS_{ATM}$) and underlying surface ($ABS_{SUR}$):

$$T = 100\% \times F_{BOA}^{\downarrow} \big/ F_{TOA}^{\downarrow}, \quad A = 100\% \times F_{TOA}^{\uparrow} \big/ F_{TOA}^{\downarrow}, \tag{4}$$

$$ABS_{ATM} = 100\% \times \left(F_{TOA}^{net} - F_{BOA}^{net}\right) \big/ F_{TOA}^{\downarrow},$$

$$ABS_{SUR} = 100\% \times F_{BOA}^{net} \big/ F_{TOA}^{\downarrow}.$$

Here, $F_{BOA(TOA)}^{\downarrow(\uparrow)}$ are broadband solar radiative fluxes in the interval of (0.2-5 µm), the symbols $\downarrow(\uparrow)$ are used
to denote the downward and upward radiation at the top of the atmosphere (TOA) and bottom of the
atmosphere (BOA). Radiative influxes $F^{net}$ at different levels are calculated as follows

$$F_{BOA(TOA)}^{net} = F_{BOA(TOA)}^{\downarrow} - F_{BOA(TOA)}^{\uparrow} \tag{5}$$

The results of radiative flux simulation were used to calculate the direct radiative effect (DRE) at TOA
and BOA and in atmospheric column:

$$\Phi_{TOA(BOA)} = F_{TOA(BOA)}^{net,a} - F_{TOA(BOA)}^{net,R}, \quad \Phi_{ATM} = \Phi_{TOA} - \Phi_{BOA}, \tag{6}$$

where the superscripts "$a$" and "$R$" correspond to the calculations in aerosol-molecular atmosphere and in the
aerosol–free atmosphere, with only molecular (Rayleigh) scattering and absorption taken into consideration.
The negative and positive DRE values are associated with an aerosol cooling and warming, both at TOA and
BOA. In addition to the proper DRE, we also considered the radiative effect efficiency

$$\Phi_{TOA(BOA)}^{e} = \Phi_{TOA(BOA)} \big/ \tau_{550}, \quad \Phi_{ATM}^{e} = \Phi_{ATM} \big/ \tau_{550}, \tag{7}$$

which characterizes the rate at which the atmosphere is forced per unit of AOD at 550 nm at TOA and BOA.
The $\Phi^{e}$ value depends on reflective properties of underlying surface, size distribution of aerosol particles, and
their chemical composition; also, it depends on AOD, to some extent (due to the multiple scattering effects).

**5.1 Model and input data**






The broadband fluxes of the solar radiation in the molecular-aerosol plane-parallel atmosphere were calculated
using the algorithm of the Monte Carlo method, which we developed earlier (Zhuravleva et al., 2009a). The
radiative fluxes at a given atmospheric level z are represented as a sum of fluxes in separate wavelength
intervals:

$$F^{\downarrow(\uparrow)}(z) = \sum_{i=1}^{M} F_i^{\downarrow(\uparrow)}(z), \tag{8}$$

where M =31 is the number of bands $\Delta\lambda = (\lambda_i, \lambda_{i+1})$, $i = 1,...M - 1$, $\lambda_1 = 0.2\ \mu m$, $\lambda_M = 5.0\ \mu m$. Within each
subinterval, the optical characteristics of aerosol and molecular scattering coefficient are assumed to be
constant and equal to their values in the middle of subinterval. The transmission function is approximated by
a finite exponential series (k-distribution method). The algorithm intrinsically takes into account the multiple
scattering, absorption by aerosol and molecular particles, as well as reflection of incident radiation from the
underlying surface according to Lambert law. The comparisons showed that the numerical simulation results
are in a satisfactory agreement with results of line-by-line calculations and data of field measurements
(Tvorogov et al., 2008; Zhuravleva et al., 2009a, 2014).
The spectral AOD (340-1020 nm), the water vapor content, the column-integrated single scattering
albedo, and the asymmetry parameter taken from AERONET were used as the main input parameters of the
algorithm under the smoke conditions. In the interval of 440-1020 nm, the spectral values $\omega_\lambda$ and $g_\lambda$ have
been linearly interpolated from the values of SSA and AF retrieved at the four AERONET inversion
wavelengths, while for $\lambda \leq 440\ nm$ and $\lambda \geq 1020\ nm$ they have been considered constant, similar to Garsia
et al.(2012) and Panchenko et al. (2012). Under the background conditions, Level-2.0 retrieval products for
the single scattering albedo and asymmetry factor, obtained on the basis of standard AERONET algorithm, are
not available due to low AOD values ($\tau_{550} = 0.13$ according to data of multiyear ground-based measurements
under summer conditions, Sakerin et al., 2009; Sakerin and Kabanov, 2015). Therefore, OPAC model
(averaged continental aerosol, relative air humidity is 70%, (Hess et al., 1998) was used to simulate the
radiative characteristics under the conditions of the weakly turbid atmosphere.
As in (Panchenko et al., 2012), the aerosol optical depth was assumed to be constant outside the
wavelength interval of 340-1020 nm: $\tau(\lambda \leq 340\,nm) = \tau_{340}$, $\tau(\lambda \geq 1020\,nm) = \tau_{1020}$. Such an approach was
chosen for the following reasons. The contribution of solar radiation, incoming at TOA in the interval of 200-
340 nm, is about 3.5%; therefore, in the absence of measurements, the specified character of spectral
dependence of AOD cannot influence significantly the simulation results. Based on the data of multiyear
ground-based observations on the territory of Siberia, Sakerin and Kabanov (2007) showed that AOD hardly
varies in the interval λ>1000 nm; therefore, we can assume that τ(λ>1000 nm)~ τ(λ~1000 nm).
The molecular absorption coefficients were calculated on the basis of HITRAN2008 database and
MT_CKD v.2.4 continuum model (http://rtweb.aer.com/continuum_frame.html), using a regional model of
temperature, pressure, and water vapor concentration profiles (Komarov and Lomakina, 2008), taking into
account the absorption by all atmospheric gases, which was presented in the AFGL meteorological model




(Anderson et al., 1986). The profiles of ozone $O_3$ and carbon dioxide $CO_2$ were specified taking into
consideration the data of multiyear observations, obtained on the territory of the Western Siberia during
summer period: the total $O_3$ content was taken to be equal to 340 DU (according to data of TOMS satellite
instrumentation, 2000-2010), and the $CO_2$ mixing ratio was taken to be equal to 380 ppm (according to data of
aircraft sensing in 1997-2007 (Arshinov et al., 2009)).

It is well known that the surface albedo affects (primarily upward) radiative fluxes and may even cause

DRE sign reversal (see, e.g., Zhuravleva et al., 2009b; Garsia et al., 2012; Tomasi et al., 2015). The processes
of soot sedimentation on the ground and appearance of blanked patches from burning produce changes in the
surface reflectance and possible variations in the radiative characteristics of the atmosphere. A detailed
discussion of these problems is beyond the scope of the present work; therefore, all calculations below were
performed for the same surface albedos $A_{s,\lambda}$, specified using multiyear data of MODIS satellite
measurements for the region of Tomsk in June – August (Moody et al., 2005).

Data of Fontenla et al. (1999) were used to account for the spectral behavior of the solar constant.

The applicability of the algorithm and the approach to specifying the set of input parameters was

confirmed by our earlier results of the complex radiation experiments (Zhuravleva et al., 2009a, 2014).

**5.2 Simulation results**

Recent studies showed that consideration of only specific spectral range of aerosol properties and neglect of
uncertainties in specifying the input parameters (aerosol optical characteristics, surface albedo, total content
and vertical profiles of concentrations of atmospheric gases, etc.) may be important error sources in estimates
of aerosol radiation effects (Myhre et al., 2003; Zhou et al., 2005; Garcia et al., 2008, 2012; Zhuravleva et al.,
2009ab). Magnitude of these errors depends mainly on aerosol type (background continental, oceanic, biomass
burning, urban-industrial, desert dust, etc.), reflection model, and albedo of underlying surface, as well as on
solar zenith angle (see, e.g., Garsia et al., 2008, 2012, 2014; Zhuravleva and Sakerin, 2009b).

In this work, we present the *diurnally* average radiative effects for 5 different situations (Table 4). To

estimate correctly their uncertainties, it is necessary to take into account both variations in the characteristics
of the atmosphere and underlying surface during the day and the dependence of uncertainties on the solar
zenith angle. The estimates, presented in the literature, were generally obtained for fixed illumination
conditions (see, e.g., Garsia et al., 2008, 2012, 2014; Zhuravleva and Sakerin, 2009b), while uncertainty
estimates for *diurnally* average radiative effects are scarce (see, e.g., Tomasi et al., 2015, Esteve et., al., 2016).
In this work, we restrict ourselves to a discussion of just radiation effects in different situations. Issue of what
is the magnitude of uncertainty due to inaccurate information on input parameters of the problem needs
additional numerical experiments and is beyond the scope of the present work.

Average values of AOD, W, SSA, and AF, which reflect the "average" radiative effects of smoke and

background aerosol, were chosen as input parameters for the first three cases. In order to avoid the effect of
astronomical factor, the instantaneous values of $F_{TOA(BOA)}^{\downarrow(\uparrow)}$ were calculated for period between sunrise and





sunset on July 15 for the latitude of Tomsk (56° N). Cases 4 and 5 correspond to the conditions of strong (July,
27) and relatively weak (July, 14) aerosol turbidity and, as such, make it possible to estimate the variability
range of radiative characteristics in the period of smoke mist.
Figure 12 shows how radiation characteristics of the atmosphere and underlying surface are
redistributed under different conditions.
The atmospheric transmittance (and, hence, $ABS_{SUR}$) was maximal (71%) under the background
conditions. The appearance of optically dense smoke cloud had a consequence that $T$ decreased to 60% (Case
3). Immediately in the period of 2012 smoke mist, the $T$ value varied from 67% (July 14) to 37% (July 27),
being almost a factor of two smaller in the latter case than under the background conditions. Data of
calculations agree with results of measurements of total radiative fluxes in Tomsk (Sklyadneva et al., 2015),
which indicated that on July 27, 2012 the total radiation decreased by about 50% relative to the usual
conditions. Atmospheric albedo was also maximal (34%) on this day.
In analysis of $ABS_{ATM}$, the existence of two competing factors should be taken into consideration. If
we represent the atmospheric absorption as a series in the order of scattering, its $n$th term will be proportional
to $\omega_\lambda^{n-1}(1-\omega_\lambda)$; and for large $\omega_\lambda$ the contribution of high orders of scattering will be significant. Therefore,
AOD increase (and, consequently, increment in the average order of scattering) favors absorption growth due
to the contribution of high orders of scattering. At the same time, large SSA values act as a factor reducing the
atmospheric absorptance.
The $ABS_{ATM}$ value, calculated with averaged parameters, increased from 22% (Case 1) to 26%
(Case 3); while for certain situations during summer 2012 the atmospheric absorptance varied in the range of
24-36%. A substantial increment in $ABS_{ATM}$ on July 27 had been a consequence of a considerable (almost an
order of magnitude) increase in AOD, accompanied by growth of the average order of scattering.
The direct radiative effect of aerosol for fixed characteristics of underlying surface and fixed solar
zenith angle primarily depends on AOD, single scattering albedo, and asymmetry factor (Zhou et al., 2005;
Yu et al., 2006; Tomasi et al., 2015). The results, presented in Fig. 12b, show a cooling effect of aerosol at the
top and bottom of the atmosphere. As expected, the interrelation between DRE values was determined by
aerosol optical depth and had been maximal (in absolute value) on July 27, 2012: $\Phi_{BOA} = -150\ \mathrm{W\,m^{-2}}$,
$\Phi_{TOA} = -75\ \mathrm{W\,m^{-2}}$, $\Phi_{ATM} = 75\ \mathrm{W\,m^{-2}}$. Under the background conditions, the DRE value was minimal: -
13 W m$^{-2}$ at BOA and -5 W m$^{-2}$ at TOA.
The AOD influence on direct radiative effect efficiency $\Phi^e$, as compared to DRE, is much less
pronounced, making it possible to estimate the influence of absorbing and scattering aerosol properties on
radiative effects. However, it should be kept in mind that the increase of multiple scattering effects and
attenuation of the transmitted radiation for large AOD moderate their effect (Conant et al., 2003).
For small atmospheric turbidity, the aerosol single scattering albedo weakly influences the radiative
effect efficiency, and high $\left|\Phi^e\right|$ values at TOA, BOA, and in the atmosphere are primarily determined by AOD





(Case 1, Fig. 12c). The increase of efficiency for lowest AOD range was also noted by other authors (see, e.g.,
Garcia et al., 2012; background continental regions).
The aerosol optical depths in the ordinary smokes and on July 14 are close in value, whereas SSA
values substantially differ (Table 4). A consequence of this is the inequality $\left|\Phi^e(\text{Case 2})\right| > \left|\Phi^e(\text{Case 4})\right|$,
primarily because of higher absorptance of aerosol particles in ordinary smokes. Comparison of cases 3-5
shows that the $\left|\Phi^e(\text{Case 4})\right|$ value is maximal for SSA close to unity (Fig. 12c), suggesting that the influence
of absorbing and scattering aerosol properties on radiative effect efficiency is moderated for high AOD values.

**6 Conclusion**

Previous studies showed that smokes from vegetation burning, together with large volcanic eruptions, are most
intense natural sources of aerosol-gas emissions in boreal zones of the planet; they strongly influence the
radiation budget on the scales of large regions a few weeks long. One such event, i.e., smoke mist due to
massive forest fires, had taken place during summer 2012 in a few Siberian regions.
In this work, we present the results of complex study of the optical and microphysical characteristics
and radiation effects of aerosol, observed under the conditions of severe smoke turbidity of the atmosphere.
The basis for analysis had been photometric observations (AERONET/Tomsk-22) of spectral solar radiation
with the use of the algorithms of solution of inverse problems of atmospheric optics and model calculations of
the main components of shortwave radiation budget. The obtained radiation characteristics in smoke mist
situation are compared with data of multiyear (2003-2013) observations under the background conditions, for
"ordinary" smokes, as well as with results of studying the smoke aerosol obtained by other authors.
Intensive wildfires in 2012 in Siberia had led to extremely high aerosol loading of the atmosphere: the
average AOD(500 nm) in the period of smoke mist had been 0.95±0.86, a factor of 6 larger than under the
background conditions (0.16±0.08), and almost a factor of 2.5 larger than AOD in "ordinary" smokes
(0.36±0.18). The AOD value exceeded 3 in certain periods of measurements (on July 24-28).
Like at other AERONET sites, where smoke aerosol was recorded, in Tomsk the volume aerosol size
distributions were bimodal with dominating fine-mode fraction. In June-August 2012, the mean median radius
of fine fraction $r_v^f$ had increased to 0.18 µm as compared to "ordinary" smokes ( $r_v^f = 0.16\ \mu m$ ). The width
of the fine mode distribution increased, as was the case in period of intense fires during summer 2004-2005 in
Alaska [Eck et al., 2009]. In contrast to data of multiyear observations, in summer 2012 an increase in the
volume concentration and median radius of the coarse mode was observed with the growing AOD.
The average imaginary refractive index of the Tomsk smoke mist shows low values and relatively large
decrease in $\kappa_\lambda$ as the wavelength grows from 440 nm (0.0067) to 675 nm (0.0054). The $\kappa_\lambda$ variations in the
wavelength interval of 675-1020 m are not as strong. At the same time, in ordinary smokes the imaginary part
of refractive index showed spectral behavior close to neutral one ( $\kappa_\lambda \approx 0.01$ ).
A consequence of small values of the imaginary index of refraction coupled with the large fine-mode



particle radius had been, on the average, quite high single scattering albedos of smoke aerosol, weakly varying with wavelength (0.96). A similar situation was also observed at other AERONET sites, located in boreal zone of Eurasia, during extended wildfires: Moscow (2002 and 2010), Alaska (2004 and 2005). At the same time, increasing spectral behavior of single scattering albedo with the growing wavelength was observed in certain periods of smoke turbidities. Possibly, this spectral dependence was because "brown" carbon, which absorbs most intensely in ultraviolet spectral region, was present in the atmosphere. A comparative analysis also showed that SSA values and their spectral dependence differ between smoke mist and "ordinary" smokes. In the latter case, the increase in absorptance of aerosol particles and decrease in SSA with the growing wavelength well correspond to the character of SSA variations, recorded in boreal zone of USA and Canada according to data of multiyear observations (Dubovik et al., 2002).

Such extraordinary events as severe smoke mist (Siberia, 2012; Moscow, 2010; etc.) are quite rare in occurrence. However, when optical and microphysical characteristics, obtained in these periods of time, are included in the total dataset (ordinary smokes, usual conditions), this may introduce substantial changes in the statistical characteristics of aerosol, typical for a given region (see, e.g., Sayer et al., 2014), and will lead to a greater uncertainty in estimates of radiation-climatic effects of aerosol. In our opinion, it is more correct to use the obtained information for analysis of characteristics of "pure" smoke aerosol (in view of its predominating contribution), as well as for estimation of maximal radiation effect of smokes.

The results of simulating the diurnally average radiative characteristics, presented in the work, reflect the "average" radiative effects of smoke and background aerosol. As compared to background conditions and "ordinary" smokes, under the conditions of smoke mist the cooling effect of aerosol intensifies (predominately due to a substantial increase in AOD): direct radiative effects at the bottom and top of the atmosphere are -13, -35, and - 60 W m$^{-2}$, and -5, -14, and -35 W m$^{-2}$ respectively. Values of direct radiative effect efficiency $\Phi^e$ under the background conditions and under the conditions of ordinary smokes are comparable ($\Phi^e_{BOA}$ ~-100 W m$^{-2}$, $\Phi^e_{TOA}$ ~-40 W m$^{-2}$, $\Phi^e_{ATM}$ ~-60 W m$^{-2}$), while in smoke mist the increase in single scattering albedo leads to almost halving of $\Phi^e_{ATM}$. We note that this work presents estimates of daytime values of aerosol radiation effects under different atmospheric situations. The issue of what is the uncertainty of these estimates, caused by insufficiently exact information on the input parameters of radiation calculations, requires further study.

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





Table 1. Average (± SD) characteristics of AOD and water vapor content of the atmospherein the period of smoke mist in 2012 as compared to multiyear data for July and "ordinary" smokes (April-October 2003-2011 and 2013)

| Characteristics | July, multiyear | | Smoke mist | "Ordinary" smokes |
| --- | --- | --- | --- | --- |
| | Background conditions | Total dataset (with smokes) | (Jun 17 – Aug 6, 2012) | |
| $\tau_{340}$ | 0.25±0.11 | 0.33±0.26 | 1.37±1.10 | 0.58±0.28 |
| $\tau_{500}$ | 0.16±0.08 | 0.21±0.18 | 0.95±0.86 | 0.36±0.18 |
| $\tau_{870}$ | 0.07±0.04 | 0.09±0.08 | 0.40±0.40 | 0.15±0.09 |
| α | 1.48±0.29 | 1.49±0.28 | 1.58±0.21 | 1.59±0.22 |
| β | 0.058±0.037 | 0.074±0.064 | 0.33±0.35 | 0.12±0.071 |
| W, g/cm$^2$ | 2.09±0.56 | 2.16±0.57 | 2.19±0.58 | 1.93±0.79 |

Table 2. Average (±SD), minimal, and maximal values of AOD (550 nm) in different periods of 2012 smoke mist according to data of ground-based and satellite measurements over the central part of the Western Siberia

| Period | Satellite data | | Ground-based observations (Tomsk-22) | |
| --- | --- | --- | --- | --- |
| | Average ± SD | min/max | Average ± SD | min/max |
| June 6-10 | 0.23±0.08 | 0.04/0.41 | 0.19 ± 0.07 | 0.10/0.27 |
| July 1-4 | 1.40±0.62 | 0.18/3.25 | 1.51± 1.05 | 0.85/2.72 |
| July 10-14 | 0.28±0.11 | 0.08/0.76 | 0.22± 0.11 | 0.11/0.35 |
| July 24-28 | 1.19±1.08 | 0.00/3.63 | 2.74± 1.07 | 1.32/3.83 |



Table 3. Optical and microphysical properties of biomass burning aerosol retrieved from AERONET Tomsk sites ("Tomsk", "Tomsk-22")

| Aerosol characteristics | Smoke mist, 2012 | Ordinary smoke, 2003-2011,2013 |
|---|---|---|
| $n(440/675/870/1020)$ | 1.469/1.486/1.503/1.499 | 1.454/1.471/1.485/1.499 |
| $\kappa(440/675/870/1020)$, $(\times10^2)$ | 0.673/0.538/0.501/0.488 | 1.126/1.005/1.057/1.06 |
| $\omega(440/675/870/1020)$ | 0.96/0.96/0.95/0.95 | 0.92/0.91/0.89/0.88 |
| $g(440/675/870/1020)$ | 0.68/0.59/0.54/0.51 | 0.68/0.59/0.55/0.54 |
| $r_v^{\mathrm{f}}\,(\mu m)$, $\sigma^{\mathrm{f}}\,(\mu m)$ | 0.181±0.02; 0.507±0.058 $r_v^{\mathrm{f}}=0.166+0.017\tau_{440}$ $(R=0.47)$ | 0.161±0.030; 0.422±0.058 $r_v^{\mathrm{f}}=0.153+0.011\tau_{440}$ $(R=0.13)$ |
| $r_v^{\mathrm{c}}\,(\mu m)$, $\sigma^{\mathrm{c}}\,(\mu m)$ | 3.321±0.365; 0.703±0.056 $r_v^{\mathrm{c}}=2.928+0.44\tau_{440}$ $(R=0.66)$ | 2.911±0.665; 0.694±0.077 $r_v^{\mathrm{c}}=3.10-0.249\tau_{440}$ $(R=-0.13)$ |
| $C_v^{\mathrm{f}}\left(\mu m^3/\mu m^2\right)$ | 0.11±0.065 $C_v^{\mathrm{f}}=0.009+0.114\tau_{440}$ $(R=0.96)$ | 0.097±0.042 $C_v^{\mathrm{f}}=0.017+0.106\tau_{440}$ $(R=0.87)$ |
| $C_v^{\mathrm{c}}\left(\mu m^3/\mu m^2\right)$ | 0.025±0.012 $C_v^{\mathrm{c}}=0.013+0.014\tau_{440}$ $(R=0.62)$ | 0.062±0.04 $C_v^{\mathrm{c}}=0.036+0.035\tau_{440}$ $(R=0.24)$ |

Table 4. Parameters of radiation calculations

| | Case | $\tau_{550}$ | $\alpha_{440-870}$ | W | $\omega_{550}$ | $g_{550}$ |
|---|---|---|---|---|---|---|
| 1 | Background aerosol | 0.13 | 1.46 | 2.1 | 0.925 | 0.7 |
| 2 | "Ordinary" smokes | 0.33 | 1.59 | 1.9 | 0.92 | 0.64 |
| 3 | Smoke mist, 2012 | 0.84 | 1.58 | 2.2 | 0.96 | 0.64 |
| 4 | July 14 | 0.34 | 1.74 | 2.5 | 0.98 | 0.64 |
| 5 | July 27 | 3.54 | 1.54 | 1.2 | 0.94 | 0.65 |
| Spectral surface albedo | $A_{s,470}=0.036$; $A_{s,560}=0.067$; $A_{s,670}=0.068$; $A_{s,870}=0.251$; $A_{s,1250}=0.285$; $A_{s,1650}=0.196$; $A_{s,2150}=0.1$ | | | | | |




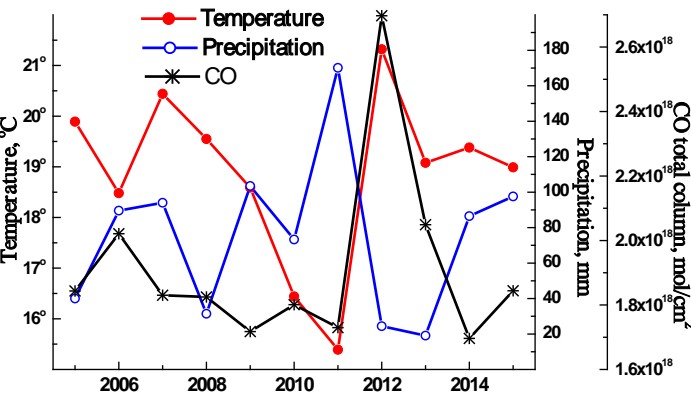

Figure 1. Interannual variations in July-average temperatures and precipitation amount according to data from "Tomsk" meteorological station, as well as interannual variations in total carbon monoxide content according to satellite observations over the territory of Tomsk region (http://giovanni.sci.gsfc.nasa.gov/).

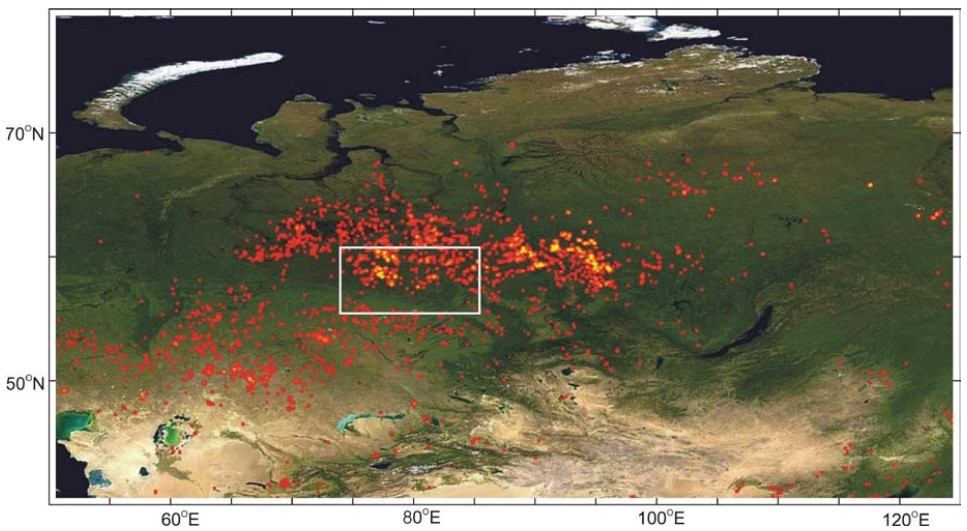

Figure 2. The map of forest fires over Siberia in the period from July 19 to 28, 2012 (http://lance-modis.eosdis.nasa.gov/cgi-bin/imagery/firemaps.cgi). The Tomsk region (56-61ºN; 75-88ºE) is highlighted by box.





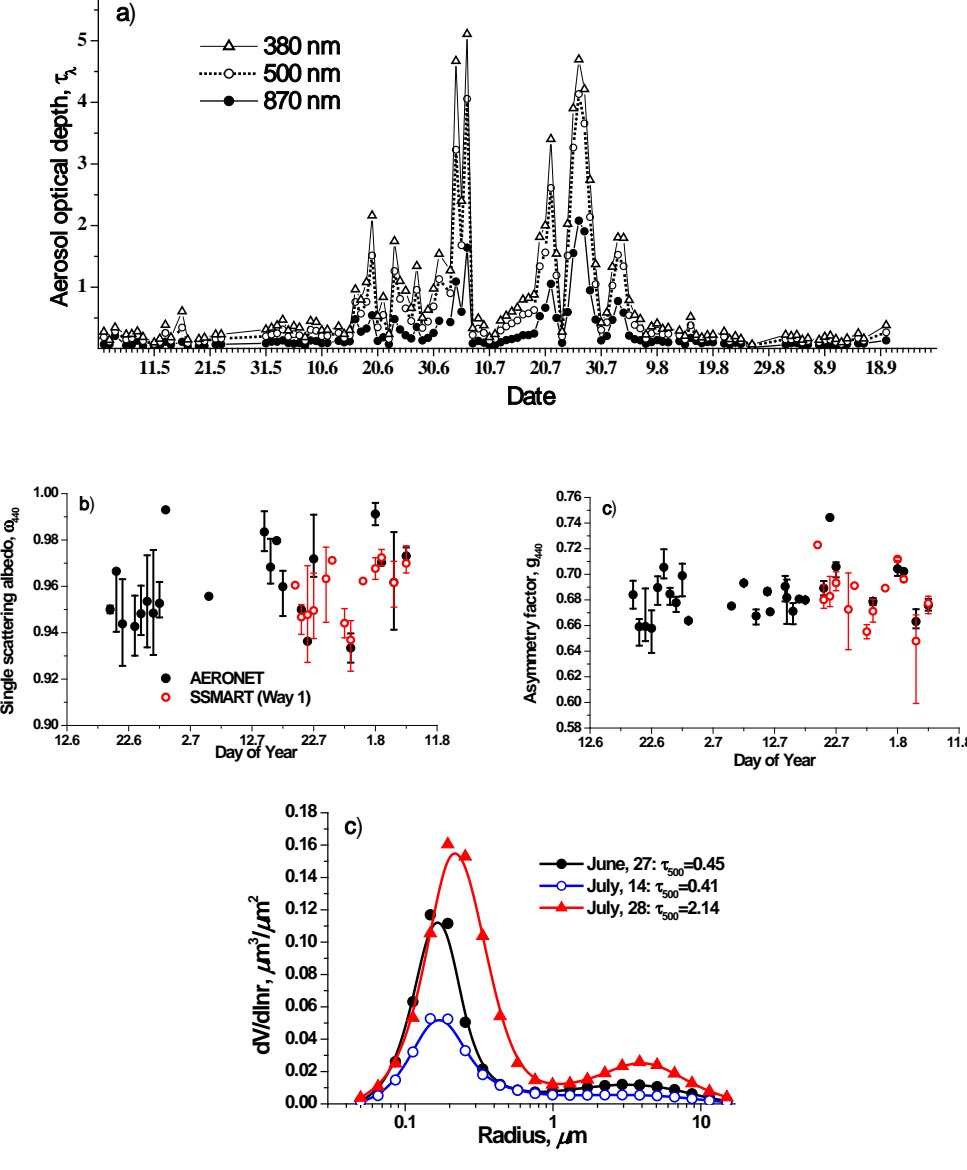

Figure 3. (a) Daily average AOD, (b) single scattering albedo and (c) asymmetry factor time series, and (d) examples of particle size distribution in "Tomsk-22" during summer period of 2012.





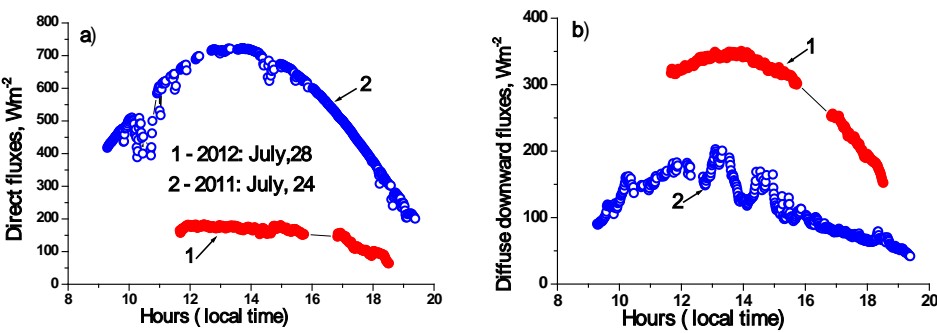

Figure 4. Daytime behavior of (a) direct and (b) diffuse fluxes of solar radiation under the usual conditions (July 24, 2011) and during smoke mist (July 28, 2012) in the region of "Fonovaya" observatory.

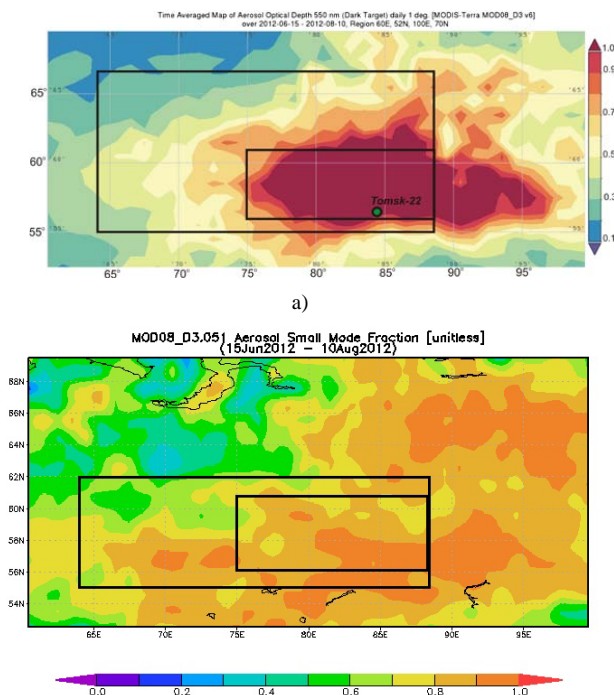

Figure 5. Spatial distributions of (a) AOD (MODIS collection 6) and (b) Aerosol Small Mode Fraction (MODIS collection 5) over Western and Eastern Siberia during June 15 – August 10, 2012. The central part of the Western Siberia (55-62ºN; 64-88ºE) and the Tomsk region (56-61ºN; 75-88ºE) are highlighted by boxes.



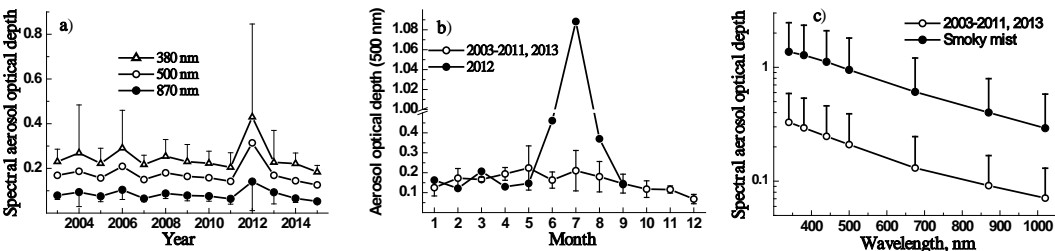

Figure 6. (a) Multiyear averages of spectral aerosol optical depth; (b) monthly averages of $\tau_{500}$ in 2012 compared with the multiyear averages for the period of 2003-2011, 2013; and (c) average spectral AOD dependences in the period of smoke mist as compared to multiyear data in July (Tomsk, 2003-2011, 2013, combined dataset of smoke and background situations)

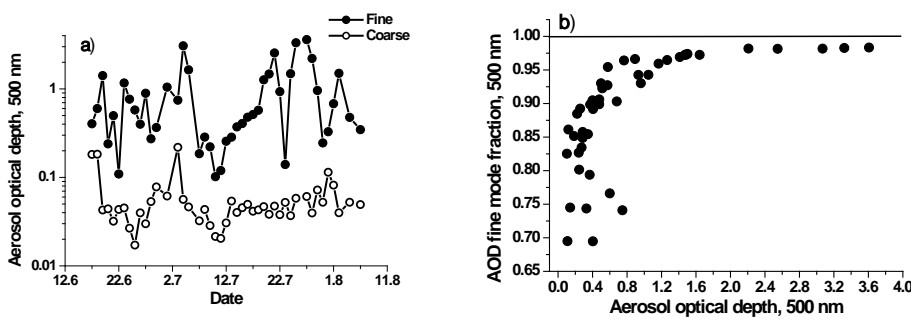

Figure 7. (a) Time series of computed average fine and coarse AOD mode; (b) fine mode fraction of AOD as a function of AOD (500 nm) for the Tomsk data during smoke mist in 2012





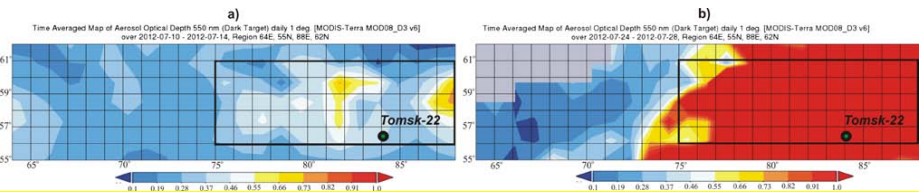

Figure 8. Spatial AOD distribution over the central part of the Western Siberia according to MODIS data for summer 2012: a) July 10-14, b) July 24-28.




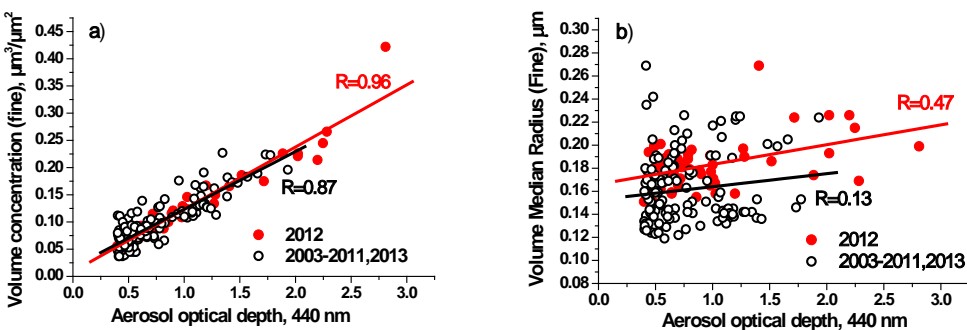

Figure 9. (a) Fine mode volume concentration and (b) median radius versus AOD(440 nm) in Tomsk.

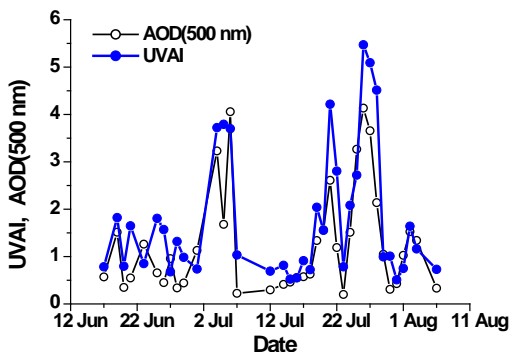

Figure 10. Ultraviolet Aerosol Index (http://giovanni.sci.gsfc.nasa.gov/giovanni/) near Tomsk (55-57°N, 83-

85°E) and AOD according to data of ground-based measurements (http://aeronet.gsfc.nasa.gov)





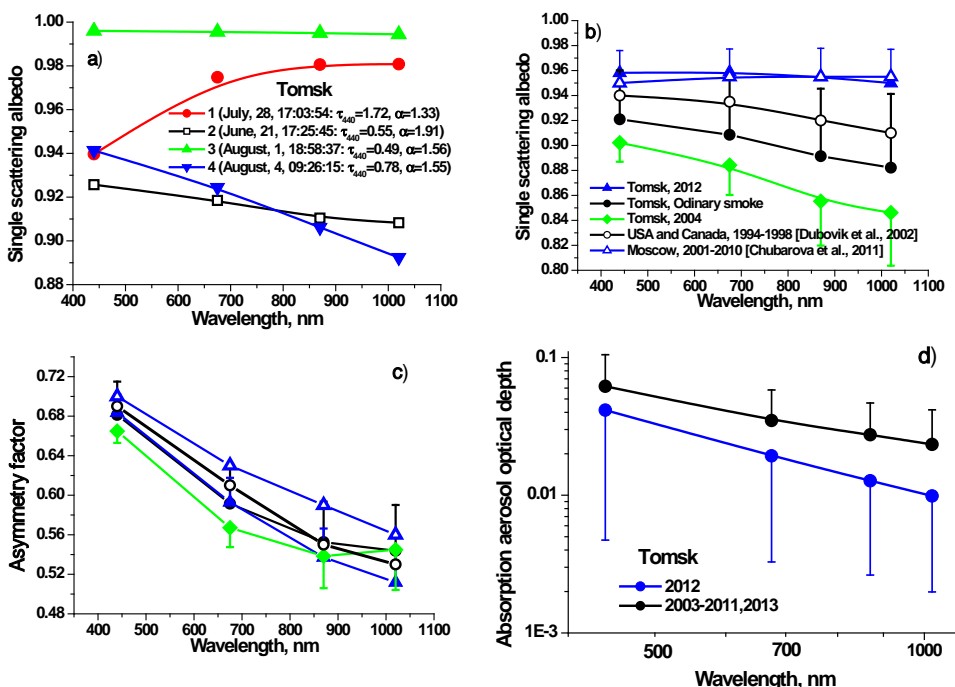

Figure 11. (a) Examples of spectral dependence of SSA; averaged (b) SSA and (c) AF for different periods and regions of observations; and (d) average spectral absorption aerosol optical depth for 2012 smoke mist and "ordinary" smokes.



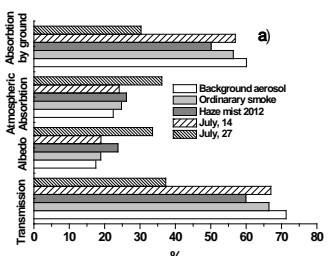

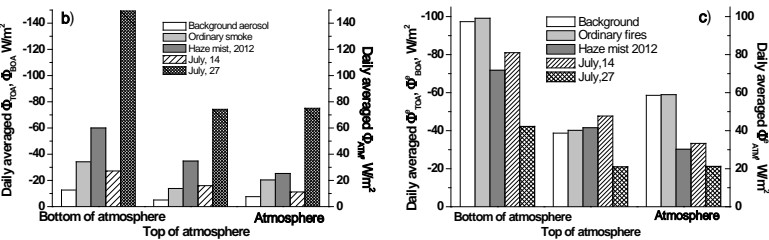

Figure 12. Radiative effects of aerosol under different atmospheric conditions.