# Peer review of "Radiative characteristics of aerosol under smoke mist conditions in Siberia 1 during summer 2012 2 3 Tatiana B. Zhuravleva1, Dmitriy M. Kabanov1, Ilmir M. Nasrtdinov1, Tatiana V. Russkova1, 4 Sergey M. Sakerin1, Alexander Smirnov2,3</s"

_Atmospheric Measurement Techniques, 2016_

## Referee Comment (RC1) · Anonymous Referee #1 · 12 Sep 2016

Summary The paper is devoted to the analysis of microphysical, optical and radiative properties of smoke aerosol during extreme fire events in summer of 2012 over Western Siberia. The data from Tomsk AERONET site and satellite observations were involved into consideration. It was revealed that intensive wildfires in 2012 in Siberia were characterized by extremely high aerosol loading. The average AOD at 500 in the period of smoke mist has been six times greater than under the background conditions, and almost a factor of 2.5 larger than mean AOD in "ordinary" smokes in other years. The AOD value reached 5 in certain periods (on July 24-28). Smoke aerosol size distributions had bimodal structure with prevailing fine fraction. Abnormal atmospheric turbidity and high values of single scattering albedo resulted in strong cooling effect of

smoke aerosol. The results, presented in the manuscript, are of considerable interest from the point of view of radiative and climatic effects of the large-scale intensive wild-fires in the boreal zones. The paper of T.B. Zhuravleva et al. is worthy of publication in AMT once the general and specific comments have been addressed.

General comments 1. It was stated in Chapter 2.1 that "An original approach, relying upon ground-based spectral measurements of AOD and radiance phase functions, was also used in addition to the algorithm of Dubovik and King (2000) to solve the inverse problem" (lines 104-105). But throughout the paper, only AERONET retrievals are considered. It would be worth to compare them with SSMART retrievals in certain situations. Especially it concerns SSA and asymmetry factors retrieved directly from the data of AOD and sky radiance measurements by means of RTE equation solution. 2. Relative contribution of coarse aerosol fraction into AOD is much greater in IR spectral region (1020 nm), than in blue (440 nm). In Table 3, only the regression equation between volume concentration of coarse aerosol and AOD (440 nm) is given. What about 1020 nm? Specific comments: Line 24. -"SSA(440 nm)=0,92". Change comma to dot. Lines 33-35. -"The maximal values of DRE were observed on July 27 (AOD(500 nm)=3.5), when DRE(BOA) reached -180 W m-2, while DRE(TOA) and DRE of the atmosphere were -80 W m-2." These values do not coincide with those on lines 560 – 561. Line38. - "diurnally radiative effects of smoke and background aerosol". Diurnal. Line 82. - "CE 318 has been operated at "Fonovaya" observatory…". Words "has been" are extra. Lines 177-178. - "(particle size 0.4-15 um)". Do these values refer to particle radius? Lines 253-255. -"The slight increase in Ångström exponent $\alpha$, which depends on the interrelation between contributions of fine and coarse aerosols to AOD, also indicates that small particles predominate in smoke aerosol". Angstrom exponent depends not only upon relation between fine and coarse fraction, but also upon parameters of fine fraction. Lines 596-597. -"The width of the fine mode distribution increased…". What are the quantitative parameters of the fine mode broadening? Lines 601-602. -"The $\kappa\lambda$ variations in the wavelength interval of 675-1020 m are not as strong". 675 – 1020 nm. May be "not so strong"? Fine fraction is called throughout the

text as "finely dispersed fraction".

---

## Referee Comment (RC2) · Anonymous Referee #3 · 24 Oct 2016

This paper performs an analysis of aerosol properties at a Siberian site during the strong summer 2012 fire season, with additional comparison to background and smoke-laden periods during other years at this site. The main focus of this analysis is AERONET data, and some calculations of the radiative effects of the smoke are also made. This paper is topically relevant to ACP. I agree with the comments posted by the other reviewer, and have some additional comments of my own. I recommend revisions and another round of peer review. These comments are as follows:

1. I find the term 'smoke mist' unusual and had not encountered it before. Is this a term in common use (perhaps translated from Russian) or a new term? Perhaps I missed it. I would recommend giving a clear definition of what 'smoke mist' means, if it has a

[Figure]

specific technical meaning. The term 'mist' to me has connotations of a liquid water fog, but I don't think that is what is being talked about here. If 'smoke mist' is not a technical term then perhaps something else can be used as shorthand for the strong burning period of summer 2012.

2. As the other reviewer also noted, the paper introduces SSMART as an alternative to the AERONET data processing. However SSMART results are barely mentioned after that (there is a little in Figure 3 but that's about all). So either SSMARTS description should be removed, or SSMART results should be added and discussed in more depth. My preference is for this second option (i.e. add the SSMART results to the analysis and discussion).

3. MODIS small mode fraction. The authors show maps of MODIS Dark Target AOD from the latest Collection 6, and MODIS aerosol small mode fraction from the older version Collection 5. Small mode fraction was deleted from Collection 6 because it was found not to have any skill, and the developers recommend not to use this product. So showing maps of this does not really support the authors' discussion about the large scale structure of the aerosol during this period because the data set is so unreliable that it was discontinued by the people who created it. The MODIS small mode fraction should therefore be removed from the paper. If the authors want to show the regional pattern of total AOD and the amount from the fine mode, there are other options. The MODIS Deep Blue product gives both AOD and Angstrom exponent over land. The MISR product also has these, and I believe also has other quantities to get more directly at fine mode contribution to AOD. MISR has a narrow swath but since this is a seasonal composite, that would probably not matter too much. Either (or both) of these would be more appropriate and more convincing to use than the MODIS Dark Target data shown here.

4. On the topic of the regional extent of smoke, there are two other AERONET sites in this region which look like they may also have sampled the summer 2012 intense smoke period (based on an examination of time series from the AERONET website).

These are Irkutsk and Yekatarinburg. Tomsk is basically in the middle of these two. It would be instructive to see whether the AERONET inversions at these additional sites are similar to those at Tomsk or not (in terms of e.g. size distribution and SSA), as this will be another way to look at spatial/temporal variability in aerosol properties for this large-scale smoke event.

5. I am intrigued by the spectral shape of SSA seen on July 28 (Figure 11, red line) where the SSA shows the opposite spectral shape to the other sites. In fact that spectral behavior is opposite from what is seen for most types of smoke aerosol. The authors note that this is unusual and suspect that it may indicate an enhanced component of brown carbon compared to normal. However there is no specific evidence for this idea. I think it would be good to dig deeper and see if the reasons for this can be found with more confidence. Since the authors include several members of the AERONET team, perhaps they can take a closer look at this case and see whether there is any indication of a retrieval problem or if it is probably real. I went on the AERONET website and found that this unusual spectral pattern of SSA was found for about one week at the end of July, with more usual patterns before and after. Perhaps HYSPLIT back trajectories or some other method will reveal something extra about possible contributions to the aerosol observed during this week as opposed to at other times.

6. Radiation calculations and discussion: I understand, if I have read correctly here, that the calculations of the smoke aerosol radiative effects here are diurnally-averaged. But I wonder if some instantaneous calculations could also be included. This would allow comparison with for example the diurnal cycle of observed radiative fluxes shown in Figure 4. CERES data could also be used as a point of comparison, as I believe this includes various flux products. In a more general sense, I think it would be good to find some way to make this information useful for other studies (as a reader I am not sure what I would use these numbers for in my own research). Perhaps something like forcing efficiency (i.e. flux change per unit AOD) could be calculated, to provide a point of comparison with aerosols in other regions.

7. UV AOD observations. As a minor point, I note that the Tomsk sun photometer includes UV bands. I believe that the SDA product (for AERONET fine/coarse AOD) also uses these to provide information about the spectral curvature of the Angstrom exponent. Table 1 shows that the visible Angstrom exponent for the summer 2012 cases was similar to that of normal smoke. Table 3 shows that the peak radius and spread for summer 2012, though, were somewhat larger than normal smoke. I would expect this difference to be reflected in the UV behavior of the AOD (either the curvature of the Angstrom exponent, or just in the Angstrom exponent over the 340-500 nm range). Perhaps this information could also be added to Table 1 as another comparison between the two periods. The calculated absorption Angstrom exponents could also be added here and discussed.

8. Figure 6, could the one-sided error bars be described in the caption for panels a and c? My guess is that these represent the mean and maximum values? Or is this a plotting error? Same question for panel d of Figure 11.

I am sorry for the delay in providing this review; I was invited to provide a review later on in the review process and then was away on travel until recently.

---

## Author Comment (AC1) · 21 Nov 2016

We thank the reviewer for taking the time to review our paper manuscript and provide useful feedback. Our responses are provided below.

General comments 1. It was stated in Chapter 2.1 that "An original approach, relying upon ground-based spectral measurements of AOD and radiance phase functions, was also used in addition to the algorithm of Dubovik and King (2000) to solve the inverse problem" (lines 104-105). But throughout the paper, only AERONET retrievals are considered. It would be worth to compare them with SSMART retrievals in certain situations. Especially it concerns SSA and asymmetry factors retrieved directly from the data of AOD and sky radiance measurements by means of RTE equation solution.

[Figure]

We accounted for this comment in the paper text. Subsection 4.2.3 is extended: we extracted a separate subsection 4.2.3.1, which presents comparisons of single scattering albedo and asymmetry factor, retrieved using SSMART and AERONET algorithms.

2. Relative contribution of coarse aerosol fraction into AOD is much greater in IR spectral region (1020 nm), than in blue (440 nm). In Table 3, only the regression equation between volume concentration of coarse aerosol and AOD (440 nm) is given. What about 1020 nm?

Naturally, the relative contribution of coarse aerosol fraction into AOD is much greater in IR spectral region. We present regression equations between concentration of coarse aerosol and AOD(1020). The correlation coefficient between and for the sample of "ordinary smokes, 2003-2011, 2013" is almost two times higher than for ; whereas the correlation coefficients R(CVc,tau440) and R(CVc,tau1020) during summer 2012 were practically the same (see a fragment of the table 1). We think it is inexpedient to introduce these data in the paper text. Specific comments

Line 24. -"SSA(440 nm)=0,92". Change comma to dot. This is corrected in the text

Lines 33-35. -"The maximal values of DRE were observed on July 27 (AOD(500 nm)=3.5), when DRE(BOA) reached -180Wm-2, while DRE(TOA) and DRE of the atmosphere were -80 W m-2." These values do not coincide with those on lines 560 – 561. We agree with this comment. Changes are introduced in the abstract: "The maximal values of DRE were observed on July 27 (AOD(500 nm)=3.5), when DRE(BOA) reached -150 W m(-2), while DRE(TOA) and DRE of the atmosphere were -75 W m(-2)."

Line38. - "diurnally radiative effects of smoke and background aerosol". Diurnal. We agree: "diurnal radiative effects of smoke and background aerosol" is more correct

Line 82. - "CE 318 has been operated at "Fonovaya" observatory:" Words "has been" are extra. This is corrected in the text.

Lines 177-178. - "(particle size 0.4-15 micrometer)". Do these values refer to particle radius? These values refer to particle diameter. This is corrected in the text: "(particle diameter 0.25-30 micrometer)"

Line 253-255. -"The slight increase in Ångström exponent, which depends on the interrelation between contributions of fine and coarse aerosols to AOD, also indicates that small particles predominate in smoke aerosol". Angstrom exponent depends not only upon relation between fine and coarse fraction, but also upon parameters of fine fraction.

We did not write that the Ångström exponent depends ONLY on the interrelation between contributions of fine and coarse aerosols. The dependence on the particle refractive index and sizes is also known to exist. However, the interrelation between two aerosol fractions has the main effect on the ïĄą value. To avoid ambiguous interpretation, the word "mainly" is added in the following sentence: "The slight increase in Ångström exponent, which mainly depends on the interrelation between contributions of fine and coarse aerosols to AOD, also indicates that small particles predominate in smoke aerosol."

Lines 596-597. -"The width of the fine mode distribution increased...". What are the quantitative parameters of the fine mode broadening? In addition to the volume median radius, Table 3 also presents the standard deviation for fine fraction, which characterizes the width of the fine mode distribution. This value varies from 0.422±0.058 under the conditions of the ordinary smokes to 0.507±0.058 during summer 2012.

Lines 601-602. "The . . . variations in the wavelength interval of 675-1020 nm are not as strong". May be "not so strong"? This is corrected in the text

Fine fraction is called throughout the text as "finely dispersed fraction" This is corrected in the text. Phrase of "finely dispersed fraction" is changed to "fine fraction"

Please also note the supplement to this comment:

http://www.atmos-meas-tech-discuss.net/amt-2016-244/amt-2016-244-AC1-supplement.pdf

[Figure]

**Supplement:**

Table 1.

| Aerosol characteristics | Extreme smokes, 2012 | Ordinary smokes, 2003-2011, 2013 |
|---|---|---|
| $C_v^c \left( \mu m^3 / \mu m^2 \right)$ | 0.025±0.012 $C_v^c = 0.013 + 0.014\tau_{440}$ (R=0.62) | 0.062±0.04 $C_v^c = 0.036 + 0.035\tau_{440}$ (R=0.24) |
| $C_v^c \left( \mu m^3 / \mu m^2 \right)$ | 0.025±0.012 $C_v^c = 0.014 + 0.055\tau_{1020}$ (R=0.63) | 0.062±0.04 $C_v^c = 0.017 + 0.233\tau_{1020}$ (R=0.45) |

---

## Author Comment (AC2) · 21 Nov 2016

Interactive comment on "Radiative characteristics of aerosol under smoke mist conditions in Siberia during summer 2012" by T. Zhuravleva et al. (Referee 2)

We thank the reviewer for taking the time to review our paper and provide useful feedback. Our responses are provided below.

1. I find the term 'smoke mist' unusual and had not encountered it before. Is this a term in common use (perhaps translated from Russian) or a new term? Perhaps I missed it. I would recommend giving a clear definition of what 'smoke mist' means, if it has a specific technical meaning. The term 'mist' to me has connotations of a liquid water

[Figure]

fog, but I don't think that is what is being talked about here. If 'smoke mist' is not a technical term then perhaps something else can be used as shorthand for the strong burning period of summer 2012.

We agree with the reviewer: the term of "smoke mist" is not quite a correct translation from Russian. Title of the paper is changed. The new version of the title is as follows: "Radiative characteristics of aerosol during extreme fire event over Siberia in summer 2012". The phrases of "extreme" smokes (by analogy to "ordinary" smokes, severe forest fires, etc.) are used instead of the term of "smoke mist" in the paper text.

2. As the other reviewer also noted, the paper introduces SSMART as an alternative to the AERONET data processing. However SSMART results are barely mentioned after that (there is a little in Figure 3 but that's about all). So either SSMARTS description should be removed, or SSMART results should be added and discussed in more depth. My preference is for this second option (i.e. add the SSMART results to the analysis and discussion).

We accounted for this comment in the paper text. Subsection 4.2.3 is extended: we extracted a separate subsection 4.2.3.1, which presents comparisons of single scattering albedo and asymmetry factor, retrieved using SSMART and AERONET algorithms.

3. MODIS small mode fraction. The authors show maps of MODIS Dark Target AOD from the latest Collection 6, and MODIS aerosol small mode fraction from the older version Collection 5. Small mode fraction was deleted from Collection 6 because it was found not to have any skill, and the developers recommend not to use this product. So showing maps of this does not really support the authors' discussion about the large scale structure of the aerosol during this period because the data set is so unreliable that it was discontinued by the people who created it. The MODIS small mode fraction should therefore be removed from the paper. If the authors want to show the regional pattern of total AOD and the amount from the fine mode, there are other options. The MODIS Deep Blue product gives both AOD and Angstrom exponent over land. The

MISR product also has these, and I believe also has other quantities to get more directly at fine mode contribution to AOD. MISR has a narrow swath but since this is a seasonal composite, that would probably not matter too much. Either (or both) of these would be more appropriate and more convincing to use than the MODIS Dark Target data shown here.

We agree with this comment. The map of MODIS aerosol small mode fraction from the older version Collection 5 was removed from the paper.

4. On the topic of the regional extent of smoke, there are two other AERONET sites in this region which look like they may also have sampled the summer 2012 intense smoke period (based on an examination of time series from the AERONET website).

The notion of "region" is relative, to a sufficient degree. The Siberian region spans about 3.5 thousand kilometers, of which 2000 km is occupied by the Western Siberia and 1.5 thousand kilometers by the Eastern Siberia. The distance between "neighboring" AERONET stations is also very large: "Tomsk" and "Yekaterinburg" are separated by 1500 km, and "Tomsk" and "Irkutsk" are separated by 1300 km. In addition, the "Irkutsk" station is actually not in proper Irkutsk, but, rather, 100 km away from the city in mountain valley (Tory settlement with its own microclimate). Taking into account these distances, the regional-scale smokes in different cases may be observed at two to three sites, or just at one. We also note that "local" smokes from closer lying burning sources may also be observed at each site. In the specific case (July-August 2012), the main forest fire seats were heavily concentrated in the north of the Western and Eastern Siberia, and their effect was significantly manifested in Tomsk region. The "Yekaterinburg" and "Irkutsk" stations turned out to be on the periphery of these severe smokes, and AOD remained practically unaffected at these sites (see Figure 1). We also note that the AOD increase in the region of "Irkutsk" in the second decade of July was most probably due to local fires. Taking into account the evidences above, we feel no sense in a comparative analysis with aerosol characteristics at "Yekaterinburg" and "Irkutsk" stations.

5. I am intrigued by the spectral shape of SSA seen on July 28 (Figure 11, red line) where the SSA shows the opposite spectral shape to the other sites. In fact that spectral behavior is opposite from what is seen for most types of smoke aerosol. The authors note that this is unusual and suspect that it may indicate an enhanced component of brown carbon compared to normal. However there is no specific evidence for this idea. I think it would be good to dig deeper and see if the reasons for this can be found with more confidence. Since the authors include several members of the AERONET team, perhaps they can take a closer look at this case and see whether there is any indication of a retrieval problem or if it is probably real. I went on the AERONET website and found that this unusual spectral pattern of SSA was found for about one week at the end of July, with more usual patterns before and after. Perhaps HYSPLIT back trajectories or some other method will reveal something extra about possible contributions to the aerosol observed during this week as opposed to at other times.

The SSA behavior on July 28, which increased with wavelength, was indeed atypical. At the same time, similar "anomalous" situations during summer 2012 numbered 14 out of 65. We also note that the revealed specific features were also observed at other AERONET sites. In particular, in the period of severe forest fires in Moscow (2010) and Alaska (Bonanza Creek, 2004), there were situations when SSA values increased with wavelength (see Figure 2). We would like to note that on days with "anomalous" spectral dependence of the single scattering albedo spectral dependence of aerosol optical depth changed. Angstrom parameter was lower; fine mode aerosol radii was higher (Eck et al. (2009) suggested that peat fires might be responsible for this and presented similar patterns in Fig.10 and 11 of his 2009 paper.

We note that this spectral dependence of SSA during summer 2012 was also obtained by us through another approach to retrieving the optical and microphysical characteristics of aerosol (SSMART algorithm). SSMART-derived data for a number of situations are added in the paper (July, 28). In the situations, considered here, both algorithms (SSMART and AERONET) showed (qualitatively and quantitatively) consistent spectral

dependence of SSA (we added new Figure – Fig.12). As a hypothesis – and solely as a hypothesis! – we suggested that the reason was due to the manifestation of absorbing properties of brown carbon. We discussed with members of AERONET team the reasons for atypical spectral behavior of SSA, recorded in separate atmospheric situations during summer 2012 in Tomsk. In our opinion, Đř deeper understanding of the revealed fact requires additional studies, which would be based on a larger AERONET dataset and, possibly, on parallel in-situ measurements (such as measurements of aerosol chemical composition). At the present stage, we just state the presence of "anomalous" situations.

Comparison of aerosol optical characteristics, retrieved using AERONET and SSMART algorithms, as well as brief comments on a possible reason for the anomalous spectral behavior of SSA are added in the paper text (subsection 4.2.3.1. See also response to Comment 2).

6. Radiation calculations and discussion: I understand, if I have read correctly here, that the calculations of the smoke aerosol radiative effects here are diurnally-averaged. But I wonder if some instantaneous calculations could also be included. This would allow comparison with for example the diurnal cycle of observed radiative fluxes shown in Figure 4. CERES data could also be used as a point of comparison, as I believe this includes various flux products. In a more general sense, I think it would be good to find some way to make this information useful for other studies (as a reader I am not sure what I would use these numbers for in my own research). Perhaps something like forcing efficiency (i.e. flux change per unit AOD) could be calculated, to provide a point of comparison with aerosols in other regions.

The reviewer is absolutely right: data on diurnal-averaged radiative characteristics were obtained on the basis of instantaneous calculations.

As regards the comparisons of radiative flux simulations and measurements, we think it is inexpedient to include them in the contents of this paper. A comparison of simulations

and measurements for a large dataset under the background conditions (AOD(550 nm) = 0.03-0.22) was performed by us earlier during a closed radiation experiment. In our work (Zhuravleva et al. Solar radiative fluxes in the clear-sky atmosphere of Western Siberia: A comparison of simulations with field measurements, Atmos. and Oceanic Optics, March 2014, Vol. 27, pp. 176-186) we showed a satisfactory correspondence between the results of field experiments and numerical simulation. The correct interpretation of the experimental data in comparison with the numerical simulation requires a comprehensive approach. It is necessary to keep in mind a number of circumstances. In particular, for large AOD values (AOD(500 nm)>1), multiple scattering effects, the presence of clouds invisible through smoke haze, etc. may affect significantly the AOD retrieval accuracy (and, as a consequence, the results of retrieval of other optical characteristics). Comparison of calculation results and field ground-based measurements under these extreme conditions is a rather complicated problem. This is even more true for a comparison of simulation results and CERES data, as recommended by reviewer. Unfortunately, our ability in summer 2012 were limited. For this reason, the comparison task is beyond the scope of present study.

Figure 4 (the results of measurements of direct and downward scattered radiation) is presented in the paper to show how large the difference may be in the solar radiative fluxes between background situations and severe fire conditions.

As regards the possibility of using the obtained information in other studies, the data on the forcing efficiency are already present in the paper text (Fig. 13c – in a new numbering).

7. UV AOD observations. As a minor point, I note that the Tomsk sun photometer includes UV bands. I believe that the SDA product (for AERONET fine/coarse AOD) also uses these to provide information about the spectral curvature of the Angstrom exponent. Table 1 shows that the visible Angstrom exponent for the summer 2012 cases was similar to that of normal smoke. Table 3 shows that the peak radius and spread for summer 2012, though, were somewhat larger than normal smoke. I would

expect this difference to be reflected in the UV behavior of the AOD (either the curvature of the Angstrom exponent, or just in the Angstrom exponent over the 340-500 nm range). Perhaps this information could also be added to Table 1 as another comparison between the two periods. The calculated absorption Angstrom exponents could also be added here and discussed.

We are careful about using the Angstrom exponent to interpret the microphysical characteristics of aerosol.

First, the Angstrom exponent is determined with much larger uncertainty than AOD themselves, with the uncertainty rapidly increasing with shortening of wavelength interval. Of course, the value of the Angstrom exponent is most often different in particular wavelength intervals. However, explanation of these differences may lead to false conclusions due to insufficient reliability of Angstrom exponent estimates on short intervals. Second, variations in Angstrom exponent are difficult to interpret unambiguously (it depends on interrelation between fine and coarse aerosols, as well as on refractive index and distribution function of particles of fine aerosol). Without invoking to additional information, there will be always an uncertainty regarding precisely what are the factors due to which Angstrom exponent changed or did not, or these are just uncertainties of estimates.

Nonetheless, according to suggestions of the reviewer the Table 1 was revised to include the average values of Angstrom exponent in the interval of 340-500 nm.

8. Figure 6, could the one-sided error bars be described in the caption for panels a and c? My guess is that these represent the mean and maximum values? Or is this a plotting error? Same question for panel d of Figure 11.

Figures 6 and 12( in a new numbering) present standard deviations. One-sided error bars are used solely in order not to overburden the figures.

Please also note the supplement to this comment:

[Figure]

http://www.atmos-meas-tech-discuss.net/amt-2016-244/amt-2016-244-AC2-supplement.pdf

[Figure]

**Supplement:**

[Figure]

Figire 1. Time variations in atmospheric AOD (0.5 µm) at three AERONET stations in the Siberian region (July-August 2012)

Moscow, 2010

[Figure]

Bonanza Creek, 2004

Figure 2. Single scattering albedo of aerosol particles during the strong fires at Moscow and Bonanza Creek.